# H₂-driven biocatalysis for flavin-dependent ene-reduction in a continuous closed-loop flow system utilizing H₂ from water electrolysis

Guiyeoul Lim[1], Donato Calabrese[1], Allison Wolder [2], Paul R. F. Cordero[1], Dörte Rother[1,3], Florian F. Mulks [4], Caroline E. Paul [2] & Lars Lauterbach [1]✉

Despite the increasing demand for efficient and sustainable chemical processes, the development of scalable systems using biocatalysis for fine chemical production remains a significant challenge. We have developed a scalable flow system using immobilized enzymes to facilitate flavin-dependent biocatalysis, targeting as a proof-of-concept asymmetric alkene reduction. The system integrates a flavin-dependent Old Yellow Enzyme (OYE) and a soluble hydrogenase to enable H₂-driven regeneration of the OYE cofactor FMNH₂. Molecular hydrogen was produced by water electrolysis using a proton exchange membrane (PEM) electrolyzer and introduced into the flow system *via* a designed gas membrane addition module at a high diffusion rate. The flow system shows remarkable stability and reusability, consistently achieving >99% conversion of ketoisophorone to levodione. It also demonstrates versatility and selectivity in reducing various cyclic enones and can be extended to further flavin-based biocatalytic approaches and gas-dependent reactions. This electro-driven continuous flow system, therefore, has significant potential for advancing sustainable processes in fine chemical synthesis.

Continuous flow biocatalysis is increasingly recognized as a sustainable manufacturing methodology within the pharmaceutical and fine chemicals sectors[1–3]. It offers improved control over reactions, minimizes waste generation, and enhances efficient use of energy[4]. The application of such systems is not limited to conventional chemical synthesis, it can be adapted to biocatalysis, using enzymes as catalysts to enable efficient syntheses under mild conditions[5]. Biocatalysis in continuous flow has become more feasible due to the advancements in flow chemistry and enzyme immobilization techniques[6,7], harnessing the highly stereo- and regioselective catalytic power of enzymes. One of the most useful reactions for generating stereogenic centers involves the asymmetric reduction of activated C=C bonds[8,9]. This ene-reduction can be catalyzed by an ene reductase from the Old Yellow Enzyme family (OYE)[10], which contains a prosthetic flavin mononucleotide (FMN) cofactor. Through *trans*-hydrogenation, OYE catalyzes the asymmetric reduction of α,β-unsaturated carbonyl compounds[11]. The commonly recognized process of regenerating enzyme-bound FMNH₂ involves the reduced nicotinamide cofactor NAD(P)H, although OYE shows promiscuity with respect to the sources of reductant[12–14], including receiving electrons from reduced free-flavins[10,15,16]. Flavin-based biocatalysis has recently attracted considerable attention not only for its application in C=C double bond reduction, but also for its applicability in epoxidation, hydroxylation and nitro group reduction, due to its low-cost cofactor compared to NAD(P)H[17–19]. Consequently, alternative regeneration methods such as photochemical and electrochemical approaches have been reported to regenerate the free flavins. These methods, however, are constrained by limitations, such as low conversion rates and stability[20–24]. In vivo ene-reduction in continuous flow using OYEs has demonstrated high conversion rates[25,26]. However, in vivo reactions pose challenges in downstream processing, as unwanted materials or contaminants can be co-purified. In-vitro biocatalysis in flow chemistry can significantly simplify the downstream process, and the potential for scaling up flavin-based biocatalysis through the application of flow chemistry remains unexplored[4].

Molecular hydrogen (H₂) serves as an ideal reductant since it does not produce any waste and can be produced by electrolysis of water using

[1]Institute of Applied Microbiology—iAMB RWTH Aachen University, Aachen, Germany. [2]Biocatalysis Section, Department Biotechnology, Delft University of Technology, Delft, The Netherlands. [3]Institute for Bio-and Geosciences 1: Biotechnology Forschungzentrum Jülich GmbH, Jülich, Germany. [4]Institute of Organic Chemistry—iOC RWTH Aachen University, Aachen, Germany. ✉e-mail: lars.lauterbach@iamb.rwth-aachen.de

renewable energy. $H_2$ can be reversibly oxidized by the soluble hydrogenase of *Cupriavidus necator* (SH) to reduce $NAD^+$ to NADH[27]. This process involves a hydrogenase module, HoxHY, for $H_2$ oxidation and a reductase module, $HoxFUI_2$, for $NAD^+$ reduction. Due to its 100% atomic efficiency[28], $SH/H_2$ presents as an excellent system for NADH cofactor regeneration[29] compared to existing NAD(P)H-recycling systems, which often suffer from low activity or produce unwanted side products[30]. The native $NAD^+$-reducing ability of SH can be further extended to NADPH through rational mutagenesis, allowing $H_2$-driven NADPH regeneration[31]. Additionally, SH exhibits tolerance to ambient $O_2$ condition[32], making it a valuable tool for the recycling of NAD(P)H in various biocatalytic reactions for the production of amines and alcohols[33–35]. The application of hydrogenases has also been explored for regenerating synthetic analogous of NAD(P)H[14] and reduced flavin cofactors[17–19]. Coupling of SH with the thermostable OYE from *Thermus scotoductus* (*Ts*OYE)[36] and styrene monooxygenase allowed the regeneration of $FMNH_2$ for the reduction of C=C double bonds, and reduced flavin adenine dinucleotide ($FADH_2$) for the epoxidation of styrenes, respectively, in small-volume batch reactions[17]. This work shows that SH is an atom-efficient regeneration system that uses $H_2$ as electron source for regenerating not only NAD(P)H but also flavin cofactors. Compared to Hyd1 from *E. coli*[18], which lacks a reductase module, the SH demonstrates nearly two orders of magnitude higher specific activity ($5.8\ U\ mg^{-1}$) for FMN reduction[17]. This indicates a significant contribution of the reductase module of SH to free FMN binding and reduction[37,38].

The in-situ production of $H_2$ by water electrolysis effectively minimizes the risks associated with handling large quantities of explosive $H_2$. Photoelectrochemical oxidation of water has been applied to fuel enantioselective reduction of *Ts*OYE as a cathodic reaction[39]. However, it has a limited efficiency due to low current densities ($<1\ mA\ cm^{-2}$) when coupling the photocatalysts with the biocatalysts, posing challenges for upscaling the reaction[40]. The use of electrical energy to drive NADH-dependent biocatalytic processes in a flow system, using $H_2$ as a mediator, has been previously successfully demonstrated. In this process, electricity was used to split water *via* a simple two-electrode system to produce $H_2$ and $O_2$[41]. The hydrogen gas was then introduced into the flow system by liquid-to-liquid gas transfer, facilitated by gas-permeable Teflon™ AF-2400 tubing[41,42]. Additionally, a commercial continuous-flow hydrogenation reactor has been applied in flow biocatalysis, designed for optimizing $H_2$ availability which may become a relevant efficiency factor[43]. In this work, a simple and cost effective method of supplying different gases to the flow system was developed by integrating gas permeable tubing into a closed loop system. This approach allows continuous and efficient delivery of $H_2$ and reduced cofactors without damaging the biocatalysts. For energy-efficient water electrolysis, proton exchange membrane (PEM) electrolyzer provide a suitable method to produce pure $H_2$ with >60% voltage efficiency and >95% faradaic efficiency[44]. Compared to standard alkaline water electrolyzers, PEM electrolyzers can operate in high current densities, have instant current response, and can be coupled with renewable energy sources[45]. To introduce $H_2$ produced from the PEM electrolyzer *via* gas-to-liquid transfer without bubble formation, tube-in-tube methods or the use of gas-permeable tubes in a gas addition module have been selected for application in flow chemistry[46,47].

In this study, the main novelty is in development of a closed-loop flow platform for electro-driven flavin-dependent biocatalysis *via* $H_2$ as a mediator produced by a commercial PEM electrolyzer in combination with a gas addition module. Thermostable *Ts*OYE, as a reduced flavin-dependent model enzyme, was immobilised *via* coordination bonds and coupled to SH. To enable regeneration of the flavin cofactor, SH was immobilised by affinity binding and applied to the flow system. This system, which facilitates efficient gas-to-liquid $H_2$ transfer, was thoroughly evaluated for biocatalyst immobilization, system stability, reusability, and adaptability with different substrates, establishing its potential for technical-scale flavin-dependent biocatalytic reactions.

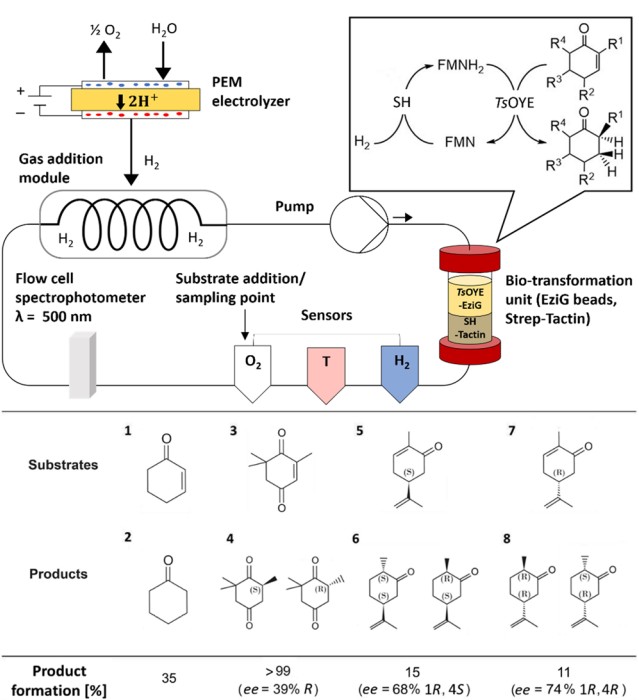

**Fig. 1 | Platform for electro-driven $FMNH_2$-dependent asymmetric reduction of cyclic enones.** $H_2$ is produced from a PEM electrolyzer using $Pt_B/Pt_C$ as hydrogen evolution catalyst. A gas permeable tubing (PVMS or Teflon) transfers the $H_2$ from the gas addition module to the flow system (17 mL, flow rate $2.6\ mL\ min^{-1}$). $H_2$ was supplied during biocatalysis to the gas addition module from the PEM electrolyzer ($H_2$ $10\ mL\ min^{-1}$, 3.4 V, 0.89 A). Clark-type sensors for $H_2$, optical sensors for $O_2$, and temperature sensors and spectrophotometer (FMN) were integrated into the flow system to monitor online the interplay between enzymes and the electrolyzer. Due to overlapping of the substrates and $FMNH_2$ absorbance peak, wavelength of 500 nm was used for FMN detection. The enzymes SH, and *Ts*OYE were immobilized by Strep-Tactin XT 4Flow (down) and EziG beads (up), respectively, and packed into a column within the flow system. Reaction conditions: 17 mL, 50 mM Tris-HCl pH 8 at 20 °C containing SH (5 mg), *Ts*OYE (6.5 mg), FMN (1 mM), catalase (5 mg) substrate **1** and **3** (25 mM), substrate **5** and **7** (5 mM). DMF as a cosolvent was added with the substrate at the ratio of 1:2. Conversions and ee values were determined by GC-FID. The flow system was at room temperature and the column with immobilized biocatalysts in temperature-controlled chamber was set at 30 °C. For upscale reaction, **3** was used at 18.5 mM concentrations.

## Results and discussion

### Gas addition module design and $H_2$ supply for $H_2$-driven flow biocatalysis

A closed-loop flow system was developed to serve as a versatile platform for electro-driven enzymatic reactions (Fig. 1). This system contained the $H_2$-dependent SH for the recycling of reduced flavin $FMNH_2$. $H_2$ was supplied from water electrolysis by a commercially available lab-scale PEM electrolyzer with an IrRuOx anode and a mixed $Pt_B/Pt_C$ (platinum black, platinum-supported carbon) cathode. The evolved $H_2$ was fed into the flow system through gas-permeable membrane tubing within a constructed metal-encased gas addition module (see SI chapter 3). For $H_2$ transfer from gas-to-liquid phase in the gas addition module, tubes made of polymethylvinylsiloxane (PVMS) with a high gas permeability and polytetrafluoroethylene (PTFE) as inert material were evaluated[41,48]. PVMS tubes with a membrane thickness of 250 μm and a length of 2 m rolled up in the gas addition module showed an $H_2$ transfer rate of $0.647\ \mu mol\ min^{-1}$ at 1 bar $H_2$ pressure in a single pass through the tubing (see SI chapter 4). Therefore, it was favored for its high gas transfer efficiency. Conversely, PTFE tubing with a lower $H_2$ transfer rate of $0.418\ \mu mol\ min^{-1}$ was used as a non-reactive alternative in cases where chemical interactions with PVMS were observed.

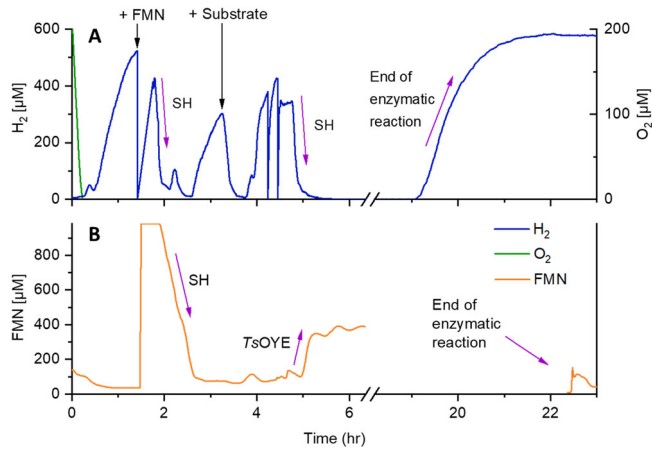

**Fig. 2 | On-line monitoring of $H_2$, $O_2$, and FMN during the electro-driven biotransformation.** The concentration of $H_2$ (A, blue), $O_2$ (A, green), and oxidized FMN (B, orange) were followed during 24 h of the transformation of **3** to **4** as exemplary biotransformation. $H_2$ was supplied to the system with the rate of 11 mL min$^{-1}$ by the PEM electrolyzer (3.4 V, 0.89 A) and gas addition module. The electrolysis was performed until $H_2$ concentration reached a plateau and then FMN was added (black arrow). After $H_2$ concentration started to increase again and FMN was fully reduced to $FMNH_2$, substrate was added to ensure reducing conditions for *Ts*OYE activity. Purple arrows indicate the activity of each enzyme: the activity of *Ts*OYE is indicated by oxidizing $FMNH_2$ to FMN, the activity of soluble hydrogenase (SH) is shown in the consumption of $H_2$, and the reduction of FMN to $FMNH_2$. The experiment was performed as described in SI 3.2.

## Closed-loop flow biocatalysis reactions with immobilized enzymes

For facilitated product isolation and allowing reusability of enzymes in the flow setup, immobilization of SH and *Ts*OYE simplifies product isolation and enables the recycling of enzymes within a flow system. However, adsorption of FMN was detected on the material Amberlite FP54™, previously used in flow setup for SH immobilization[49]. Therefore, the Strep-Tactin XT 4flow resin was assessed as a potential alternative for application in flow chemistry, aimed at Strep-tagged SH immobilization (SH-Tactin)[32,50].

Analogously, *Ts*OYE was equipped with a 6xHis-tag on the *N*-terminal site and was immobilized on porosity glass metal ion affinity EziG beads from EnginZyme[51]. Different types of EziG beads varying in hydrophilicity were evaluated for *Ts*OYE immobilization to assess performance (see SI chapter 5). Employing affinity immobilization techniques, His-tagged *Ts*OYE with EziG beads and strep-tagged SH with Strep-Tactin were used in this proof-of-concept study to rationalize the purification and immobilization of both SH and *Ts*OYE into a one-step process. This approach showed good residual activity and high immobilization yields (see SI chapter 2.4), while also simplifying the immobilization process compared to covalent bonding methods.

Subsequently, we investigated the electro-driven asymmetric reduction of cyclic enones using in-situ generated $H_2$ within the enzymatic system. Given that the $K_M$ of SH is approximately 680 µM, we used 1 mM FMN to ensure an adequate concentration close to $k_{cat}$ of SH. To avoid reactive oxygen species (ROS) from the reaction of reduced $FMNH_2$ with $O_2$ diffusing into the flow system, which potentially damages metal-dependent enzymes[52], catalase was introduced into the circulating system. The immobilized enzymes, SH and *Ts*OYE, were packed into a column which was integrated into the flow reactor. To ensure higher immobilization yield of SH, it was essential for Strep-Tactin resin to be packed and settled for 24 h before immobilization. This led to a defined arrangement of the immobilized biocatalysts in the biotransformation unit where SH-Tactin is positioned beneath the *Ts*OYE-EziG (Figs. 1, S2). Considering the relatively weaker binding affinity between the Strep-tag and Strep-Tactin compared to

coordinate bonds, there is potential for SH leaching. This involves the possibility of SH circulating around the flow setup and being reintroduced into the packed bed reactor.

During the reaction, we conducted online monitoring of various parameters, such as $H_2$ evolution and consumption, potential $O_2$ leakage, and FMN reduction (Fig. 2). This was done to monitor each catalytic step and assess whether $H_2$ concentrations and $FMNH_2$ were sufficient to initiate and progress the reaction. In that way, the functionality of each enzyme was comprehensively assessed, ensuring that no parameter within the cascade was limiting. The time at which the reaction ended could be determined by observing the saturation of $H_2$ and the low concentration of oxidised FMN at 19 and 22.5 h respectively (Fig. 2). It is worth noting that such a monitoring system, as shown in Fig. 1, had previously only been used in-vivo during fermentations and one other in flow chemistry setup[41]. This novel application in our current study represents a significant advancement in the ability to monitor enzymatic processes and identification of potential bottlenecks.

## Substrate scope of flow system

A range of substrates was chosen according to the substrate spectrum of *Ts*OYE to assess the adaptability of the flow system in 17 mL volume with various substrates (Fig. 1)[16]. Cyclohexenone (**1**) was first selected for its high water solubility and relatively high conversion[53]. In a 17 mL flow reaction, 25 mM of (**1**) yielded a 35% conversion to cyclohexanone (**2**) within 15 h, while no further conversion was detected. This conversion rate is consistent with previous batch experiments[17], but this yield is lower compared to other reactions using NADPH[53] (Table S4). The lower conversion rate could be due to the propensity of (**1**) for Michael's addition with water[54]. This combined with the potential for $FMNH_2$ to generate $H_2O_2$ in the presence of $O_2$ traces during substrate injection or sampling[32,38,55] could have contributed to side reactions and resulted to low conversions. Our detection of hydroxylated side products *via* GC-MS in the flow system suggested a potential issue with side reactions (see SI chapter 14.3). Side products were also observed when SH and *Ts*OYE were entrapped in an enzymatic membrane reactor (EMR) with a cutoff membrane of 30 kDa as an alternative immobilization technique (see SI chapter 6)[56]. (**1**) yielded a 28% conversion to (**2**) after 6 days in an EMR. While the utilization of EMR extended the enzymatic activity, it also increased the likelihood of side product formation due to its longer reaction time, rendering it unsuitable for this application.

Subsequently, the reduction of ketoisophorone (**3**) to levodione (**4**) was evaluated in the flow system due to the pharmaceutical significance of levodione as a building block. Complete conversion of 25 mM (**3**) to (**4**) was achieved 22 h after addition of substrate (>99%, Fig. 1, Figure S6). Despite constantly attaining high conversion rate, the flow reaction produced (**4**) with a relatively low optical purity, i.e., from *ee* = 77% *R* at 2 h after substrate addition to *ee* = 39% *R* at 22 h (see SI chapter 7). This decrease of optical purity has been linked to non-enzymatic racemization in water due to keto-enol tautomerization during sample preparations or to prolonged reaction conditions[17,57,58], as well as the non-enzymatic reduction of (**3**) to racemic product (**4**) by free $FMNH_2$[21]. Additionally, it has been demonstrated that a lower concentration of the enzyme contributes to higher optical purity, which introduces the possibility of reversibility of the reaction by the enzyme[57].

(*S*)- and (*R*)-carvone (**5**, **7**) were also included to assess the performance of the flow system with chiral centers. The same flow volume, 17 mL, was used with these chiral substrates (**5**, **7**), but at a reduced concentration of 5 mM due to their low solubility in aqueous solution. It is important to note that during the reaction, loss of substrate due to adsorption on PVMS tubing was detected by GC-FID. Hence, PTFE was used as the tubing for $H_2$ transfer for reactions with carvone as substrate. After 16 h substrate addition, (**5**) resulted in 15% formation of the product (1*R*, 4*S*)-dihydrocarvone (**6**) (*ee* = 68%), while (**7**) exhibited an 11% conversion to (1*R*, 4*R*)-dihydrocarvone (**8**) (*ee* = 74%) (Fig. 1). In flow reaction, substrate conversion was similar to the batch reaction[17]. However, lower optical purity was

observed, which can be also attributed to product racemization due to keto-enol tautomerization[59,60].

## Reusability of immobilized biocatalysts and upscaling

Reusability of the immobilized enzymes and the scalability of the reaction in flow setup was then investigated. Product (**4**) was synthesized from (**3**) with a maintained product formation of >99% throughout seven cycles without replacing the enzymes due to high enzyme loading (Fig. S7), demonstrating reusability of the system (see SI chapter 8). The biocatalysts utilized in these flow reactions achieved a total turnover number (TTN) of $1.01 \times 10^5$ for SH-Tactin and $1.69 \times 10^4$ for *Ts*OYE-EziG (see SI chapter 9, Table S3). Overall, the consistently high conversion rates (>99%) in multiple cycles, along with the high TTN, especially for sophisticated oxidoreductases, illustrate the remarkable stability of the biocatalysts, indicating that Strep-Tactin XT 4Flow resin and EziG beads are robust immobilization carriers for SH and *Ts*OYE, respectively. These observations corroborate previous findings, where EziG beads served as a stable enzyme carrier for imine reductase (IRED) and IRED-EziG was reused for up to 14 cycles[61]. The high TTN of SH-Tactin and *Ts*OYE-EziG highly suggests that stable enzymes have the potential to be used in upscaled flow reactions.

We also determined the Faradaic efficiency for the electro-driven reduction of (**3**) to (**4**) in the flow system. With an applied voltage of 3.3 V and a current of 500 mA, the Faradaic efficiency of the flow system was calculated to be 0.15% (see SI chapter 12). The limited electron contribution to product formation compared to other work[39] stems from the requirement of $H_2$ gas outflow from the gas addition module, which is a drawback for commercially available PEM electrolyzers that have difficulty in managing back pressure if the module remains closed. Additionally, the commercial PEM electrolyzer only facilitated stable electrolysis of water at high voltage (>3 V) to overcome the internal resistance in the cell. To increase Faradaic efficiency or the energy efficiency of the system, zero gap cells can be used where lower amount of electrical energy is used for the production of exact amount of $H_2$ that is needed for the biocatalytic reaction[62,63].

Finally, we increased the volume of the flow reaction to 185 mL to demonstrate the scalability of the system. This was done by adding 168 mL segment that increases the volume without any headspace to inhibit $O_2$ introduction to the system. Compound (**3**) was used as substrate due its high conversion in the flow setup (Fig. 1). Product formation of above 99% was achieved after 77 h (see SI chapter 10), resulting to a theoretical yield of 527 mg (**4**). The employed procedure ensured a practical product isolation. Simple extraction of the collected dispersion with diethyl ether yielded 89% (471 mg) of (**4**) in 96% purity. The TTN of mol product per mol SH-Tactin and *Ts*OYE-EziG in the 185 mL upscaled reaction reached up to $3.2 \times 10^5$ and $2.6 \times 10^4$, respectively, with indications that further upscaling can possibly yield even higher TTN. This approach slightly exceeds the performance of previous biphasic batch reactions for (**4**) production in *Ts*OYE-TTN as well as by an order of magnitude the TTN achieved by the Hyd1 hydrogenase from *E. coli*[18,64]. The high TTN values for SH-Tactin and *Ts*OYE-EziG demonstrate that immobilizing Strep-tagged SH with Strep-Tactin resin and 6xHis-tagged *Ts*OYE with EziG beads are optimal methods for significantly enhancing the robustness of biocatalysts in flavin-dependent biocatalytic reactions, surpassing the performance of previous biphasic batch reactions for (**4**) production in *Ts*OYE-TTN. Furthermore, the SH TTN during FMN recycling was an order of magnitude lower than during NADH recycling, indicating the physiological electron acceptor improves enzyme stability[41].

The specific activity of SH for reducing FMN is 5.8 U × mg$^{-1}$, with a $K_{M,FMN}$ of 680 μM[17]. Thus, we decided to use 1 mM FMN in the reaction, which resulted in a slightly lower TTN of mol product per mol FMN in comparison to other studies (see Table S4). It is worth noting that catalase (~5000 U) was added every day to mitigate potential inhibition by ROS, ensuring full product conversion. This highlights the adverse impact of ROS, particularly $H_2O_2$, on enzymatic activity. $E$ factor was also quantified to compare the waste generated by product[65,66]. 17 mL and 185 mL reactions were calculated to $E$ factor of 6.0 and 5.7, respectively (see SI chapter 13), which represents low waste production compared to other FMNH$_2$

regeneration systems[19]. This approach to electro-driven flavin-based bio-catalysis in a flow system demonstrates a promising pathway for scalable and efficient biotransformations.

## Conclusion

In this study, we established a closed-loop flow system using immobilized biocatalysts to convert electrical energy into chemical energy for production of fine chemicals. This flow system demonstrates adaptability to a range of substrates, scalability, and serves as an electro-driven system for regenerating flavin cofactors. It can be implemented to diverse biocatalytic processes that are dependent on reduced flavins such as styrene mono-oxygenase, unspecific peroxygenase-catalyzed hydroxylation *via* $H_2O_2$ from reduced flavin re-oxidation with $O_2$, and nitroreductases[10,17–19]. Apart from improving the $H_2$-availability for $H_2$-driven biocatalysis with a PEM electrolyzer and highly gas-permeable tubing PVMS, this study introduces Strep-Tactin resin and highlights EziG beads as stable enzyme carriers suitable for flavin-dependent biocatalytic applications in a flow setup. It was highly advantageous that Strep-Tactin resin did not show adsorption of FMN, unlike Amberlite resin, an alternative for immobilizing SH[49]. Additionally, using coordination bonds between the EziG beads and the 6xHis-tag on the enzyme, high residual activity was maintained while demonstrating remarkable stability, compared to other methods such as covalent irreversible immobilization. However, the use of aqueous solutions led to side reactions, notably in cyclohexenone conversion, with observed low optical purity of the chiral products. The subsequent challenge will be to incorporate immiscible organic solvents to the flow setup to increase the solubility of substrates in water and the address the poor optical purity of chiral products[57]. *Ts*OYE-EziG can be further investigated regarding its stability in organic solvents within a micro-aqueous environment, following previous studies on *Ts*OYE immobilized *via* adsorption on Celite[67].

It is also crucial to understand the significant differences in enzyme behavior between flow and batch reactions, including effects on activity, stability, and stereoselectivity, in order to further fine-tune conditions and preserve desired enzyme functionalities in flow reactions, especially in large scales. Overall, this scalable platform for electro-driven flow biocatalysis demonstrates high potential for chemical synthesis and can be flexibly adapted to other gas-dependent enzymes by attachment of corresponding gas bottles to the gas addition module. These enzymes include $O_2$-dependent P450 monooxygenase, $CH_4$-/$O_2$-dependent soluble methane mono-oxygenases, or $CO_2$-dependent formate dehydrogenases for hydroxylation and $CO_2$ fixation reactions, respectively. The integration of advanced monitoring systems, effective biocatalyst immobilization, and careful selection of reaction conditions have provided a robust basis for further investigation and optimization of enzymatic processes in flow systems.

## Methods

### Plasmid construction, growth conditions, and protein purification

*Ts*OYE (*Thermos scotoductus* SA-01 ATCC 700910) was recombinantly produced in *E. coli* BL21 Gold (DE3) cells with the corresponding plasmid in a pET-28a(+) vector, with an N-terminal His-tag (*Nde*I/*Eco*RI). For *Ts*OYE purification, 50 mL pre-culture of LB medium containing 50 μg/mL of kanamycin was inoculated with toothpick of a colony from transformed cells with plasmid pET-28a(+)-*Ts*OYE and incubated overnight at 37 °C and 180 rpm. Overexpression was carried out in 2 L of TB medium supplemented with 50 μg/mL of kanamycin. The main cultures were inoculated with 1% of pre-culture and grown at 37 °C and 180 rpm. When an $OD_{600}$ of ~0.6 was reached (~2 h 30 min), 0.1 mM of IPTG was added. After induction, cultures were incubated overnight at 30 °C and 180 rpm. Cells were harvested by centrifugation ($18{,}692 \times g$ for 30 min at 4 °C). The obtained cell pellets (~27 g wet pellet mass) were washed and suspended in approximately 30 mL MOPS-NaOH buffer (20 mM, pH 7.0), supplemented with a spatula spoon tip of DNase and MgCl$_2$, and one pill of EDTA-free cOmplete$^{TM}$ protease inhibitor cocktail. Cells were disrupted using Multi Shot Cell Disruption System (1 cycle at 1.36 kbar pressure) and

the cell debris was separated from the crude extract by centrifugation ($18,692 \times g$ for 30 min at 4 °C). The obtained supernatant was incubated in 50 mL falcon tubes in a water bath at 70 °C for 1 h 30 min. Precipitated proteins were separated by centrifugation at $38,759 \times g$ for 30 min at 4 °C. (See SI chapter 2.2 for *Ts*OYE amino acid and nucleotide sequence respectively).

The production and purification of $NAD^+$-reducing hydrogenase (SH) was performed as described in Lauterbach et al. (2013)[32].

### Immobilization of biocatalysts

Strep-Tactin® XT 4Flow resin and EziG beads were used to immobilized the enzymes. For SH immobilization, the Strep-Tactin® XT 4Flow resin was first packed inside the C 10/10 column (Cytiva) and let settle for 1 day at 4 °C. Purified SH fused with a Strep-tag was loaded into the column bed at 3.3 mg SH per g resin. For *Ts*OYE immobilization, the carrier used was EziG Amber beads. In a 50 mL centrifuge tube, *Ts*OYE fused with 6×His-tag was loaded to the carrier (14.4 mg g$^{-1}$ EziG), followed by incubation by overnight at room temperature in a roller shaker (30 rpm). The immobilized *Ts*OYE in EziG was then resuspended and added to the C 10/10 column. Catalase was added on to the top (2,000 U) before closing. The C 10/10 column, with SH-Tactin underneath and *Ts*OYE-EziG on the top, was used as the biotransformation unit and was attached to the flow setup. For further detailed steps see SI chapter 3.1.

### Reactions in flow setup

Each experiment in flow setup was done in step-reactions to check the functionality of the gas addition module, SH-Tactin, and *Ts*OYE-EziG. The flow volume (Tris-HCl 50 mM, pH 8, 30 °C) was first saturated with $H_2$ by starting the PEM electrolyzer to 0.89 A to produce rate of 11 mL min$^{-1}$ $H_2$ gas. The generated $H_2$ was added to the flow volume through the gas-addition module *via* the gas-permeable tubing (SI chapter 3). Once $H_2$ saturation was observed, 1 mL of $H_2$-saturated stock solution of FMN (concentration depending on the flow volume) was added to the flow volume to prevent $O_2$ from entering the system. After observing reduction of FMN to $FMNH_2$ and checking the functionality of SH, substrate was added (Fig. 2). The consumption of $H_2$ after free $FMNH_2$ was depleted was indicative of *Ts*OYE-EziG and SH-Tactin activities. Upon saturation of $H_2$ was observed, the reaction was stopped and the flow volume was stored for analysis.

For 17 mL flow reactions, flow volume with 1 mM FMN, 2000 U catalase, and substrate (either 25 mM ketoisophorone, 25 mM cyclohexenone, or 5 mM (*S*)- or (*R*)-carvone). For the 185 mL upscaled flow volume, the reaction involved a mobile phase with 500 μM FMN, ~5000 U catalase (added every day), and 18.5 mM of substrate ketoisophorone. A segment of 168 mL was added to the existing 17 mL to increases the volume without any headspace. For further detailed information of reaction by substrate see SI chapters 3.2 to 3.5.

### Analytics

Compound analysis was carried out on Thermo Scientific Trace GC Ultra instrument with a AS 3000 auto-sampler coupled with flame ionization detector (FID) using helium as carrier gas. Products were confirmed by reference standards. Details of the column and temperature programs used are given in Table S6. All measurements were calibrated with calibration curves. Only conversion rate was quantified with GC-FID. Enantiomeric excess (*ee* %) of compounds was assessed with Shimadzu GC-2010 equipped with an FID using helium as carrier gas (see SI chapter 14.2). Details of the column and temperature programs used are given in Table S7. Time between experiments and GC measurements was several days which also explained the poor enantiomeric excess obtained (racemization of certain products occurred). Gas chromatography-mass spectrometry was carried out on a Thermo Scientific Trace GC Ultra instrument coupled with a Thermo Scientific ISQ mass selective detector (MSD) and an additional flame ionization detector. Injector temperature: 250 °C. Split mode with a split ratio of 10. Detector temperature: 320 °C. Electron ionization of the analyte with 70 eV acceleration voltage. For further information see SI chapter 14.3. Nuclear magnetic resonance spectra were recorded on a BRUKER Avance Neo 600 spectrometer at 26 °C and were analyzed with the software Mnova 14.2.3 (Mestrelab Research). Chemical shifts δ were reported in parts per million (ppm). The residual solvent signals were used for referencing $^1H$ NMR spectra (7.26 ppm for CDCl$_3$)[68,69]. A relaxation delay of D1 = 25 s was used for quantitative $^1H$ NMR experiments. For further information see SI chapter 15.

### Reporting summary

Further information on research design is available in the Nature Portfolio Reporting Summary linked to this article.

## Data availability

Raw and metadata for this contribution are available at https://doi.org/10.5281/zenodo.10798153.

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

## Acknowledgements
We thank EnginZyme (Stockholm, Sweden) for EziG™ beads. This project has received funding from the European Union's Horizon 2020 research and innovation programme under the Marie Skłodowska-Curie grant agreement No. 955740, from the European Research Council (ERC) under the European Union's Horizon 2020 research and innovation programme (grant agreement No. 949910) and from the Deutsche Forschungsgemeinschaft (DFG, German Research Foundation) under Germany´s Excellence Strategy —Cluster of Excellence 2186 "The Fuel Science Center"—ID: 390919832. Further funding was received through a Liebig Fellowship by the Fonds der Chemischen Industrie, grant number Li 210/01 (F.F.M.).

## Author contributions
Conceptualization: L.L.; Experiments: G.L., D.C., A.W., F.F.M.; Investigation and analysis: G.L., D.C., A.W., F.M., P.R.F.C., A.D., D.R., C.E.P., L.L.; Writing—original draft: G.L., D.C., P.R.F.C.; Writing—editing and review: G.L., D.C., A.W., F.F.M., P.R.F.C., D.R., D.C.P., L.L.; Supervision and funding: L.L. All authors contributed to the discussion. All authors have given approval to the final version of the manuscript.

## Funding

## Competing interests
The authors declare no competing interests.
