## [Peer Review File · Communications Chemistry]

H₂-driven biocatalysis for flavin-dependent ene-reduction in a continuous closed-loop flow system utilizing H₂ from water electrolysisReviewers' comments:

Reviewer #1 (Remarks to the Author):

In the manuscript, Lim and colleagues describe a H₂-driven system to drive biocatalytic reduction of C=C bonds. The reactions are carried out in flow, and use hydrogen gas from water electrolysis – so they are very clean, and show good potential for application at scale. The hydrogenase/flavin-dependent reductase enzyme cascade has been previously reported, and the substrates explored here are relatively simple. However, the novelty and impact of the current paper lies in the reaction set-up, which seems to elicit a significant improvement over previous work. Furthermore, the analytical set-up (with in-line O₂ and H₂ monitoring) is especially impressive, and I can see this system being very useful for gas driven biocatalysis performed in flow in the future.

My main criticism is the confusing description of “electro-biocatalysis” in the title and throughout the manuscript. This title implies some direct communication of the enzyme with an electrode, which is misleading. For example, Vincent has published on the use of H-cube reactors for similar H₂-driven biocatalysis (DOI: 10.3389/fceng.2021.718257), and these are not described as “electro-biocatalytic” – so the description from Lim et al. risks creating a false differentiation. Similarly, the description of H₂ gas as a redox mediator, when really it could be supplied from a cylinder, seems to be unnecessarily complicated (particularly with such low faradaic efficiencies).

The paper is well written, very thorough, and the reaction engineering is interesting to read. I would recommend publication, following minor revisions:

- 1) Please correct the references to “electro-biocatalysis” in the title and elsewhere, and be clearer that this is H₂-driven biocatalysis where the hydrogen happens to derive by electrolysis – but could derive from elsewhere. The introduction of the term “BES” in the conclusion is unhelpful, as is the term “Electro/-H₂ driven biocatalysis” in the methods.
- 2) Please ensure that H-cube flow biocatalysis (e.g. 10.3389/fceng.2021.718257) is properly discussed, as it is clearly quite similar to this work
- 3) Can the authors be certain that there are no trace nicotinamide cofactors present, that may be taking part in the reaction (i.e. are there negative controls?)
- 4) Did the authors experiment with [FMN]?
- 5) The repeat reactions in Figure S7 are all run to 100%. It is not ideal to test repeatability at full conversion, as this can mask gradual deactivation.
- 6) Small typo in Table S2 – 3.21 x 10⁵ is missing superscript.

Reviewer #2 (Remarks to the Author):

The paper of Lim et al entitled “Integrated electro-biocatalysis for flavin-dependent ene-reduction in continuous flow system” investigates the possibility of integrating a flavin-dependent Old Yellow Enzyme (OYE) and a soluble hydrogenase to enable H₂-driven regeneration of the OYE cofactor FMNH₂—by producing H₂ with water electrolysis using a proton exchange membrane (PEM) electrolyzer—into a recirculating flow-through system via a designed gas membrane addition module at a high diffusion rate.

Thus, this work is a nice, worth-to-publish implementation of H₂-driven regeneration of the cofactor FMNH₂ in flow-through systems. However, there are some comments and notes related to this work.

Notes

1) The title "Integrated electro-biocatalysis for flavin-dependent ene-reduction in continuous flow system" is somewhat misleading. A continuous-flow system means in the most accepted sense a constant (in time) substrate concentration at the inlet and a constant (in time) product concentration at the outlet of the system. The system in this communication is a batch system with recirculating flow-through units—in which the product concentration is changing (in time). Therefore, I suggest indicating this in the title which may be "Integrated recirculating flow-through electro-biocatalysis for flavin-dependent ene-reduction".

2) Please mention in the introduction real continuous-flow systems using OYE enzyme-containing whole cells (e.g. Szczepańska, et al, *Sci Rep* 11, 18794 (2021). DOI:10.1038/s41598-021-97585-w) or recirculating flow-through systems with whole cell systems (e.g. Valotta, et al, *ChemSusChem* 15, e202201468 (2022). DOI: 10.1002/cssc.202201468) and compare advantages/disadvantages to those systems as well.

3) In the discussion part, a fair comparison should be made between this system versus other isolated ene-reductase systems with "traditional" NAD(P)H regeneration. It is quite visible (page 10 in SI, section 11, Table S3) that TTN for NADPH is well above the TTNs achievable with FMN.

4) In a previous study (ref 18) using [NiFe] hydrogenase 1 (Hyd1) from *Escherichia coli*, TTN over 2000 (instead of 50 in this study: Table S3) was reported for FMN in ketoisophorone reduction. What is the reason of this remarkable difference?

5) In a section (lines 213-217) of the MS it is stated: "Despite constantly attaining high conversion rate, the continuous flow reaction produced (4) with a relatively low optical purity (ee = 39% R). This decrease of optical purity was observed during the reaction time (see SI chapter 7), which is in agreement to other studies and is typically linked to racemization during sample preparations or to prolonged reaction conditions."

Because in a non-reversible enantiotopically selective biotransformation, the degree of enantiotopic selectivity ($E = k_R/k_S$, where k_S and k_R are the pseudo first-order rate constants towards the two enantiomers) should be constant, the decreasing nature of enantiomeric excess of the product in ketoisophorone reduction (Figure S16) could also reveal reversibility of the biocatalyzed process (which could explain the decreasing equilibrium conversion with increasing substrate concentration). In this aspect it would be desirable to perform control experiments to decide the reason of decreasing ee of the product in time (e.g. put a certain amount of optically active product but without the ene-reductase into the same system and check the enantiopurity over time – this could prove chemical racemization). Please check this and discuss this issue in more detail in the main manuscript!

6) A preceding study (ref 49) showed that covalently bound OYEs are more stable than their EZIG-based counterparts. Your preparative results are far away from the best outcomes of other modes of OYE usage. Why did you not select the more stable mode of immobilization (which is one obvious reason of the poorer results)?

7) Please do not use the same numbers for different compounds in the MS and in SI (side products are numbered with bold 1, 2 ... etc, which were used in the MS for denoting other compounds).

In conclusion, this work provides an alternative FMN-based flow-through cofactor regeneration system (in recirculated batch mode) for FMN-dependent enzymic reductions and thus deserves publication. However, I suggest the publication of the results after major revision addressing all issues indicated above.

Reviewer #3 (Remarks to the Author):

The study by Guiyeoul Lim and coauthors investigates the integration of electrochemical concepts into an enzymatic reaction system under flow conditions. The key parameter is a water electrolysis, which provides hydrogen to the biocatalytic reduction reaction or rather the provided hydrogenase for FMN recycling.

The study is generally a combination of existing technologies and eventually applied on a broader level in flow chemistry. The capabilities of ene reductases and OYEs in general are well investigated, immobilization is (often) considered an established technique (incl. EziG components) and the typically high potential of flow biocatalysis was presented in many studies. The addition of the PEM electrolyser module into flow conditions is the main novel concept idea. The overall concept is somehow optimized, but leaves significant potential untouched as it focuses entirely on the presentation of the overall concept, as mentioned above.

Overall comments:

- It is not really clear, why exactly this enzyme (TsOYE) was chosen as the obtained ees are limited to 39, 68 and 74%, which is below practical limits in the synthesis of chiral compounds. A different choice of a more selective enzyme would boost the entire manuscript significantly.
- The results on immobilization and characterization fills a significant amount of the entire manuscript, incl. experimental details. I'd recommend to move parts of it to the SI if the overall combination is really the overall aim of this study. Especially as Strep-Tactin resins were used for enzyme immobilization earlier in other studies.
- It is rather unclear to the reader if the chosen (very complex) process combination results in a significant improvement over other/older existing options. A few details are mentioned in table S2 in the SI, but for the key enzyme TsOYE hardly any improvement is seen. In the main text it is mentioned as "surpasses", but only to a minor effect.
- Downstream-Processing is reduced to a single sentence (extraction), which is not discussed and may not necessarily be the best option for this process (especially at lower conversions).
- The low Faradaic efficiency of 0.15% is rather low and the authors seem to mention an issue with the gas addition module, which indicates the reader that the entire concept may be considered non-optimal. This itself should have a significant effect on the E-factor.
- Is waste from catalyst preparation also part of the rough calculation of the environmental impact?

Minor issue:

- Table S2. The 5 in the SH-TTN-column is not superscript.

Response to Reviewers:

Reviewer #1 (Remarks to the Author):

In the manuscript, Lim and colleagues describe a H₂-driven system to drive biocatalytic reduction of C=C bonds. The reactions are carried out in flow, and use hydrogen gas from water electrolysis – so they are very clean, and show good potential for application at scale. The hydrogenase/flavin-dependent reductase enzyme cascade has been previously reported, and the substrates explored here are relatively simple. However, the novelty and impact of the current paper lies in the reaction set-up, which seems to elicit a significant improvement over previous work. Furthermore, the analytical set-up (with in-line O₂ and H₂ monitoring) is especially impressive, and I can see this system being very useful for gas driven biocatalysis performed in flow in the future.

Response: We would like to thank the reviewer for his/her support for the publication and also highlighting the importance of the topic.

My main criticism is the confusing description of “electro-biocatalysis” in the title and throughout the manuscript. This title implies some direct communication of the enzyme with an electrode, which is misleading. For example, Vincent has published on the use of H-cube reactors for similar H₂-driven biocatalysis (DOI: 10.3389/fceng.2021.718257), and these are not described as “electro-biocatalytic” – so the description from Lim et al. risks creating a false differentiation. Similarly, the description of H₂ gas as a redox mediator, when really it could be supplied from a cylinder, seems to be unnecessarily complicated (particularly with such low faradaic efficiencies).

The paper is well written, very thorough, and the reaction engineering is interesting to read. I would recommend publication, following minor revisions:

1) Please correct the references to “electro-biocatalysis” in the title and elsewhere, and be clearer that this is H₂-driven biocatalysis where the hydrogen happens to derive by electrolysis – but could derive from elsewhere. The introduction of the term “BES” in the conclusion is unhelpful, as is the term “Electro/-H₂ driven biocatalysis” in the methods.

Response: We thank the reviewer for this comment. To address the point clearly that the H₂ is derived from electrolysis of water separately and used for H₂-driven biocatalysis, we have changed our title to “H₂-driven biocatalysis for flavin-dependent ene-reduction in a continuous closed-loop flow system utilizing H₂ from water electrolysis”. Furthermore, we have omitted the term “BES” in the conclusion and changed the “Electro /-H₂ driven biocatalysis” in methods to “Reactions in flow setup.” These changes are highlighted in the manuscript.

2) Please ensure that H-cube flow biocatalysis (e.g. 10.3389/fceng.2021.718257) is properly discussed, as it is clearly quite similar to this work

Response: We thank the reviewer for this feedback. We agree that we should address this publication which is similar to our work. We added the following sentences in the introduction (page 2, highlighted in yellow): “Additionally, a commercial continuous-flow hydrogenation reactor has been applied in flow biocatalysis, designed for optimizing H₂ availability which may become a relevant efficiency factor [REF 10.3389/fceng.2021.718257]”.

3) Can the authors be certain that there are no trace nicotinamide cofactors present, that may be taking part in the reaction (i.e. are there negative controls?)

Response: No nicotinamide cofactors were added during the reaction. Negative control experiments without FMN were conducted in 2 mL batch reaction with no detectable conversion.

4) Did the authors experiment with [FMN]?

Response: Oxidized FMN was added in the flow system reaction, while no other flavin cofactors, such as FAD, were selected for this experiment. In order to address the reviewer comment, we stated in the manuscript, page 7, highlighted in yellow: “The specific activity of SH for reducing FMN is 5.8 U mg⁻¹ (20.3 s⁻¹), with a $K_{M,FMN}$ of 680 μM [REF DOI: 10.1039/d0cc03229h]. Thus, we decided to use 1 mM FMN in the reaction.” Higher concentrations of FMN were not applied as they were out of the linear range of the photometer integrated into the flow system.

5) The repeat reactions in Figure S7 are all run to 100%. It is not ideal to test repeatability at full conversion, as this can mask gradual deactivation.

Response: We thank the reviewer for this comment and agree that the consecutive full conversion after 19h at each cycle may not accurately reflect the deactivation rate of the immobilized biocatalysts. The high conversion is due to high amount of biocatalysts that were immobilized in the flow setup, optimized for a substrate concentration of 25 mM to complete the reaction in 18-21 hours. Moreover, the repeated reactions highlight that the immobilization carriers significantly enhance the stability and robustness of the biocatalysts for this reaction, achieving seven consecutive full conversions at high substrate concentrations. We modified the main text in the result section (page 6, highlighted in yellow) accordingly: “Product (4) was synthesised from (3) with a maintained product formation of >99% throughout seven cycles without replacing the enzymes due to high enzyme loading (Figure S7), demonstrating reusability of the system (see SI chapter 8).”

6) Small typo in Table S2 – 3.21 x 10⁵ is missing superscript.

Response: This has been revised

Reviewer #2 (Remarks to the Author):

The paper of Lim et al entitled “Integrated electro-biocatalysis for flavin-dependent ene-reduction in continuous flow system” investigates the possibility of integrating a flavin-dependent Old Yellow Enzyme (OYE) and a soluble hydrogenase to enable H₂-driven regeneration of the OYE cofactor FMNH₂—by producing H₂ with water electrolysis using a proton exchange membrane (PEM) electrolyzer—into a recirculating flow-through system via a designed gas membrane addition module at a high diffusion rate.

Thus, this work is a nice, worth-to-publish implementation of H₂-driven regeneration of the cofactor FMNH₂ in flow-through systems. However, there are some comments and notes related to this work.

Notes

1) The title “Integrated electro-biocatalysis for flavin-dependent ene-reduction in continuous flow system” is somewhat misleading. A continuous-flow system means in the most accepted sense a constant (in time) substrate concentration at the inlet and a constant (in time) product concentration at the outlet of the system. The system in this communication is a batch system with recirculating flow-through units—in which the product concentration is changing (in time). Therefore, I suggest indicating this in the title which may be “Integrated recirculating flow-through electro-biocatalysis for flavin-dependent ene-reduction”.

Response: We agree with the reviewer that continuous-flow is most accepted as a sense of a flow with an inlet and outlet of the system. In using the term "continuous-flow," it was our intention to refer to the continuous addition of reactants to a reactor, which significantly differentiates this process from a batch reaction. One such example that used “continuous-flow” in a similar context is the literature you have recommended in point 2) by Valotta et al (ChemSusChem 15, e202201468 (2022). DOI: 10.1002/cssc.202201468). However, we acknowledge that the term may be unclear, given the more accepted terminology. To address the reviewer’s concern more effectively, we changed to using the term “closed-loop” throughout our manuscript (highlighted in yellow), as used by Poznansky et al. (Frontiers in Chemical Engineering, 2021, DOI: 10.3389/fceng.2021.718257), De Santis et al. (Reaction Chemistry & Engineering, 2020, DOI: 10.1039/D0RE00335B) and Benítez-Mateos et al. (Royal Society of Chemistry, 2021, DOI:10.1039/d0re00483a) to describe a flow system with no inlet or outlet. Additionally, we have revised our title to “H₂-driven biocatalysis for flavin-dependent ene-reduction in a continuous closed-loop flow system utilizing H₂ from water electrolysis.”

2) Please mention in the introduction real continuous-flow systems using OYE enzyme-containing whole cells (e.g. Szczepańska, et al, Sci Rep 11, 18794 (2021). DOI:10.1038/s41598-021-97585-w) or recirculating flow-through systems with whole cell systems (e.g. Valotta, et al, ChemSusChem 15, e202201468 (2022). DOI: 10.1002/cssc.202201468) and compare advantages/disadvantages to those systems as well.

Response: We thank the reviewer for this comment. We have added the mentioned studies to the introduction by adding these sentences (page 1-2, highlighted in yellow) “*In-vivo* ene-reduction in continuous flow using OYEs has demonstrated high conversion rates. However, *in-vivo* reactions pose challenges in downstream processing, as unwanted materials or contaminants can be co-purified. *In-vitro* biocatalysis in flow chemistry can significantly simplify the downstream process, and the potential for scaling up flavin-based biocatalysis through the application of flow chemistry remains unexplored (Joshua et al. Royal Society of Chemistry, 2018, DOI: [10.1039/C7CS00906B](https://doi.org/10.1039/C7CS00906B))”.

3) In the discussion part, a fair comparison should be made between this system versus other isolated ene-reductase systems with “traditional” NAD(P)H regeneration. It is quite visible (page 10 in SI, section 11, Table S3) that TTN for NADPH is well above the TTNs achievable with FMN.

Response: TTNs in SI, section 11, Table S4 (prior version S3) refers to the FMN total turnover number (cofactor-TTN) not the enzyme total turnover number. Thus we added to the table S4 description (SI page 11, highlighted in yellow) “Total turnover number for the cofactors of mol product per mol cofactor” The purpose of the table was to give a comprehensive comparison of similar reactions with a different concentrations of the cofactor. Accordingly to the reviewer’s suggestion, we discussed TTN

of corresponding reaction in the manuscript (page 7, highlighted in yellow): “This approach slightly exceeds the performance of previous biphasic batch reactions for (4) production in TsOYE-TTN as well as by an order of magnitude the TTN achieved by the Hyd1 hydrogenase from *E. coli*. The high TTN values for SH-Tactin and TsOYE-EziG demonstrate that immobilizing strep-tagged SH with Strep-Tactin resin and 6xHis-tagged TsOYE with EziG beads are optimal methods for significantly enhancing the robustness of biocatalysts in flavin-dependent biocatalytic reactions, surpassing the performance of previous biphasic batch reactions for (4) production in TsOYE-TTN. Furthermore, the SH TTN during FMN recycling was an order of magnitude lower than during NADH recycling, indicating the physiological electron acceptor improves enzyme stability. The specific activity of SH for reducing FMN is $5.8 \text{ U} \times \text{mg}^{-1}$, with a $K_{M,\text{FMN}}$ of $680 \text{ } \mu\text{M}$ [REF DOI: 10.1039/d0cc03229h]. Thus, we decided to use 1 mM FMN in the reaction, which resulted in a slightly lower TTN of mol product per mol FMN in comparison to corresponding studies (see Table S4).”

4) In a previous study (ref 18) using [NiFe] hydrogenase 1 (Hyd1) from *Escherichia coli*, TTN over 2000 (instead of 50 in this study: Table S3) was reported for FMN in ketoisophorone reduction. What is the reason of this remarkable difference?

Response: We thank the reviewer for clearing this out as it could have been misleading to some readers. In fact, the SH-TTN is 3.2×10^5 , while the Hyd1-TTN is 2.0×10^4 , one order of magnitude higher. We added this in the main text, see above point 3) and in Table S4.

As pointed in point 3), we have revised Table S4 TTN to Cofactor-TTN to address reviewer’s feedback. We would like to mention that we miscalculated the Cofactor-TTN for upscale reaction, so we changed from 50 to 37 in Table S4, since 0.5 mM FMN was used for 18.5 mM substrate ketoisophorone. Also in Table S4, we added the previous study (ref 18) that the reviewer mentioned.

5) In a section (lines 213-217) of the MS it is stated: “Despite constantly attaining high conversion rate, the continuous flow reaction produced (4) with a relatively low optical purity (ee = 39% R). This decrease of optical purity was observed during the reaction time (see SI chapter 7), which is in agreement to other studies and is typically linked to racemization during sample preparations or to prolonged reaction conditions.” Because in a non-reversible enantiotopically selective biotransformation, the degree of enantiotopic selectivity ($E = k_R/k_S$, where k_S and k_R are the pseudo first-order rate constants towards the two enantiomers) should be constant, the decreasing nature of enantiomeric excess of the product in ketoisophorone reduction (Figure S16) could also reveal reversibility of the biocatalyzed process (which could explain the decreasing equilibrium conversion with increasing substrate concentration). In this aspect it would be desirable to perform control experiments to decide the reason of decreasing ee of the product in time (e.g. put a certain amount of optically active product but without the ene-reductase into the same system and check the enantiopurity over time – this could prove chemical racemization). Please check this and discuss this issue in more detail in the main manuscript!

Response: Thank you for your detailed comment. The primary reason for the lower ee is the racemization of the product caused by keto-enol tautomerization in the aqueous phase during the reaction. This has been already thoroughly studied, where Fryszkowska et al (ref 57, *Advanced synthesis and catalysis*, 2009, DOI: 10.1002/adsc.200900574) have showed that the lower optical purity is due to non-enzymatic racemization. We have addressed the reviewer’s concern by revising the sentence in the result (page 6, highlighted in yellow): “This decrease of optical purity was observed during the reaction time (see SI chapter 7), which is in agreement to other studies and is typically

linked to non-enzymatic racemization due to keto-enol tautomerization during sample preparations or to prolonged reaction conditions.”

6) A preceding study (ref 49) showed that covalently bound OYEs are more stable than their EZiG-based counterparts. Your preparative results are far away from the best outcomes of other modes of OYE usage. Why did you not select the more stable mode of immobilization (which is one obvious reason of the poorer results)?

Response: Covalently bound OYEs can be more stable as in the strength of the bond compared to coordination bonds but the residual activity after immobilization is also a factor we investigated. The preceding study (ref 49) or most of the covalent bond that they use for enzyme immobilization is done by creating imine bonds with the lysine residue of enzyme molecule and the aldehyde group. Therefore, the covalent bonding process involves modifying the enzyme molecules, which can either deactivate or significantly reduce their activity after immobilization, depending on the enzyme. EziG beads utilize coordination bonds between the nickel ions and the imidazole groups of 6xHis tags attached to the protein. This approach requires no modification of the enzyme, preserving its original residual activity and offers stronger binding compared to adsorption or other non-covalent methods. Our results on the reusability of immobilized biocatalysts (Figures S7) and the upscaled experiment (Figures S8) achieving high TTN indicate that immobilization methods employing coordination bonds are highly effective for this reaction.

7) Please do not use the same numbers for different compounds in the MS and in SI (side products are numbered with bold 1, 2 ... etc, which were used in the MS for denoting other compounds).

Response: This has been corrected. The side products are numbered as 2a, 2b, and 2c.

In conclusion, this work provides an alternative FMN-based flow-through cofactor regeneration system (in recirculated batch mode) for FMN-dependent enzymic reductions and thus deserves publication. However, I suggest the publication of the results after major revision addressing all issues indicated above.

Reviewer #3 (Remarks to the Author):

The study by Guiyeoul Lim and coauthors investigates the integration of electrochemical concepts into an enzymatic reaction system under flow conditions. The key parameter is a water electrolysis, which provides hydrogen to the biocatalytic reduction reaction or rather the provided hydrogenase for FMN recycling.

The study is generally a combination of existing technologies and eventually applied on a broader level in flow chemistry. The capabilities of ene reductases and OYEs in general are well investigated, immobilization is (often) considered an established technique (incl. EziG components) and the typically high potential of flow biocatalysis was presented in many studies. The addition of the PEM electrolyser module into flow conditions is the main novel concept idea. The overall concept is somehow optimized, but leaves significant potential untouched as it focuses entirely on the presentation of the overall concept, as mentioned above.

Overall comments:

- It is not really clear, why exactly this enzyme (TsOYE) was chosen as the obtained ees are limited to 39, 68 and 74%, which is below practical limits in the synthesis of chiral compounds. A different choice of a more selective enzyme would boost the entire manuscript significantly.

Response: The primary reason for the lower ee is the racemization of the product caused by keto-enol tautomerization in the aqueous phase during the reaction. This has been already thoroughly studied, where Fryszkowska et al (ref 57, *Advanced synthesis and catalysis*, 2009, DOI: 10.1002/adsc.200900574) have showed that the lower optical purity is due to non-enzymatic racemization. Additionally, we selected TsOYE because its thermostability makes it easy to handle and purify, allowing heat purification. High thermostability minimizes the risk of activity loss during handling. The novelty also lies in the enhanced stability achieved through coordination bond immobilization of TsOYE, a new progress towards thermophilic enzyme applications. We have integrated the reviewer's comment by adding the sentence at the end of the introduction (page 2, highlighted in yellow): "TsOYE was coupled with SH as an oxidoreductase for its high thermostability, which minimizes activity loss during handling and allows easy heat purification."

- The results on immobilization and characterization fills a significant amount of the entire manuscript, incl. experimental details. I'd recommend to move parts of it to the SI if the overall combination is really the overall aim of this study. Especially as Strep-Tactin resins were used for enzyme immobilization earlier in other studies.

Response: To incorporate reviewer's comment, parts including Table 1. of the characterization of immobilized biocatalysts were moved to the SI chapter 2.4.

- It is rather unclear to the reader if the chosen (very complex) process combination results in a significant improvement over other/older existing option. A few details are mentioned in table S2 in the SI, but for the key enzyme TsOYE hardly any improvement is seen. In the main text it is mentioned as "surpasses", but only to a minor effect.

Response: The main novelty of this system lies in its use of electricity to fuel the reactions and the potential to incorporate gases other than H₂ for flavin-dependent biocatalysis. Additionally, the system's use of coordination bonds with His-tagged TsOYE for immobilization proves highly effective, leading to higher TTN. Also, Strep-tagged SH immobilized with Strep-tactin also demonstrates good compatibility with flavin-dependent reactions, whereas adsorption methods are unsuitable due to FMN cofactor adsorption. The impact of this paper resides in the combination of these findings and the fact that it can open door to other flavin-dependent reactions.

- Downstream-Processing is reduced to a single sentence (extraction), which is not discussed and may not necessarily be the best option for this process (especially at lower conversions).

Response: *In-vitro* flow biocatalysis with immobilized enzymes offers reduced downstream processing effort. In our case, no additional purification steps were needed other than extraction and isolation due to the full conversion of the reaction to levodione. The extraction process is well studied in this study by Wenkert et al. (1973, <https://doi.org/10.1021/jo00987a026>). Extraction was done only after the reaction was finished, not as a flow-extraction.

- The low Faradaic efficiency of 0.15% is rather low and the authors seem to mention an issue with the gas addition module, which indicates the reader that the entire concept may be considered non-optimal. This itself should have a significant effect on the E-factor.

Response: Thank you for this comment. We accept that optimization is needed for H₂ production and availability. The main reason for the low faradaic efficiency is that the PEM electrolyzer was commercially bought and was only able to start electrolysis of water over 3.4 V where the production of H₂ could not be decreased. Also, the gas addition module could not be closed off in the outlet because the PEM electrolyzer is unable to withstand the back pressure. Part of our project concept is the ability to easily implement any commercial electrolyzer into the gas addition module. This allows for the easy incorporation of various gases besides H₂ without requiring complex equipment such as H-cube for gas-dependent flow reactions. To incorporate reviewer's comment, we have added the following sentences in the introduction (page 2, highlighted in yellow): "Additionally, a commercial continuous-flow hydrogenation reactor have been applied in flow biocatalysis, designed for optimizing H₂ availability which may become a relevant efficiency factor[Ref: <https://doi.org/10.3389/fceng.2021.718257>]. In this work, for the task of supplying various gases to the flow system, simple and cost-effective method by integrating gas permeable tubing in a closed loop system was designed."

Referring to the E-factor calculations, we have applied the same "simple" E-factor calculation that was used in Ramirez *et al.* (Frontiers in Catalysis, 2022, <https://doi.org/10.3389/fctls.2022.906694>), where solvent and H₂ were not counted. Also Zhao *et al.* (Chemical Science, 2021, <https://doi.org/10.1039/D1SC00295C>) and Poznansky *et al.* (Frontiers in Chemical Engineering, 2021, <https://doi.org/10.3389/fceng.2021.718257>) did not incorporate the whole mass of H₂ used during the experiment.

- Is waste from catalyst preparation also part of the rough calculation of the environmental impact?

Response: We appreciate the reviewer's comment. We believe that calculating biocatalyst preparation is an aspect of life-cycle assessment analysis, whereas the E-factor calculation focuses on the mass required for the reaction and the resulting product. We would like to refer to the three publications (Frontiers in Catalysis, 2022, <https://doi.org/10.3389/fctls.2022.906694>) (Chemical Science, 2021, <https://doi.org/10.1039/D1SC00295C>), (Frontiers in Chemical Engineering, 2021, <https://doi.org/10.3389/fceng.2021.718257>) where the biocatalyst preparation process was also not calculated. Although catalyst preparation was not included in our E-factor calculations, we can improve future preparations of catalysts to minimize waste (Organic Process Research & Development, 2010, <https://pubs.acs.org/doi/full/10.1021/op1002165>).

Minor issue:

- Table S2. The 5 in the SH-TTN-column is not superscript.

Response: This has been revised

REVIEWERS' COMMENTS:

Reviewer #1 (Remarks to the Author):

The authors have given careful consideration to the reviewer comments and have revised the manuscript appropriately. I strongly recommend publication at this stage.

Reviewer #2 (Remarks to the Author):

The revised paper of Lim et al entitled "H₂-driven biocatalysis for flavin-dependent ene-reduction in a continuous closed-loop flow system utilizing H₂ from water electrolysis" investigates the possibility of integrating a flavin-dependent Old Yellow Enzyme (OYE) and a soluble hydrogenase to enable H₂-driven regeneration of the OYE cofactor FMN-H₂—by producing H₂ with water electrolysis using a proton exchange membrane (PEM) electrolyzer—into a recirculating flow-through system via a designed gas membrane addition module at a high diffusion rate.

This work is a nice, worth-to-publish implementation of H₂-driven regeneration of the cofactor FMN-H₂ in flow-through systems. Most issues which were raised by the reviewers were properly addressed. Some minor issues, however, were not fully resolved.

Notes

5) In a section (lines 213-217) of the MS it is stated: "Despite constantly attaining high conversion rate, the continuous flow reaction produced (4) with a relatively low optical purity (ee = 39% R). This decrease of optical purity was observed during the reaction time (see SI chapter 7), which is in agreement to other studies and is typically linked to racemization during sample preparations or to prolonged reaction conditions."

Because in a non-reversible enantiotopically selective biotransformation, the degree of enantiotopic selectivity ($E = k_R/k_S$, where k_S and k_R are the pseudo first-order rate constants towards the two enantiomers) should be constant, the decreasing nature of enantiomeric excess of the product in ketoisophorone reduction (Figure S16) could also reveal reversibility of the biocatalyzed process (which could explain the decreasing equilibrium conversion with increasing substrate concentration). In this aspect it would be desirable to perform control experiments to decide the reason of decreasing ee of the product in time (e.g. put a certain amount of optically active product but without the ene-reductase into the same system and check the enantiopurity over time – this could prove chemical racemization). Please check this and discuss this issue in more detail in the main manuscript!

Response: Thank you for your detailed comment. The primary reason for the lower ee is the racemization of the product caused by keto-enol tautomerization in the aqueous phase during the reaction. This has been already thoroughly studied, where Fryszkowska et al (ref 57, *Advanced synthesis and catalysis*, 2009, DOI: 10.1002/adsc.200900574) have showed that the lower optical purity is due to non-enzymatic racemization. We have addressed the reviewer's concern by revising the sentence in the result (page 6, highlighted in yellow): "This decrease of optical purity was observed during the reaction time (see SI chapter 7), which is in agreement to other studies and is typically linked to non-enzymatic racemization due to keto-enol tautomerization during sample preparations or to prolonged reaction conditions."

Extended note: In the cited work [Fryszkowska et al (ref 57, *Advanced synthesis and catalysis*, 2009, DOI: 10.1002/adsc.200900574)], the Table 5 shows the ee of the results of ketoisophorone formation under the same conditions and same reaction time as a function of enzyme concentration. In fact, the increasing enzyme concentration resulted in decreasing ee, which cannot be rationalized by the racemization (same pH, same time) but is in line with the lowering degree of reversibility (due to lower amount of enzyme). Please mention the reversibility of the reaction as another major reason of lowering the ee of the product in the discussion.

6) A preceding study (ref 49) showed that covalently bound OYEs are more stable than their EZIG-based counterparts. Your preparative results are far away from the best outcomes of other modes

of OYE usage. Why did you not select the more stable mode of immobilization (which is one obvious reason of the poorer results)?

Response: Covalently bound OYEs can be more stable as in the strength of the bond compared to coordination bonds but the residual activity after immobilization is also a factor we investigated. The preceding study (ref 49) or most of the covalent bond that they use for enzyme immobilization is done by creating imine bonds with the lysine residue of enzyme molecule and the aldehyde group. Therefore, the covalent bonding process involves modifying the enzyme molecules, which can either deactivate or significantly reduce their activity after immobilization, depending on the enzyme. EziG beads utilize coordination bonds between the nickel ions and the imidazole groups of 6xHis tags attached to the protein. This approach requires no modification of the enzyme, preserving its original residual activity and offers stronger binding compared to adsorption or other non-covalent methods. Our results on the reusability of immobilized biocatalysts (Figures S7) and the upscaled experiment (Figures S8) achieving high TTN indicate that immobilization methods employing coordination bonds are highly effective for this reaction.

Extended note: Your response is not a reasonable scientific answer. If you can report (and compare) the immobilization yields (Y_i : mass of immobilized enzyme/mass of total amount of enzyme $\times 100$) and the activity yields (Y_a : activity of immobilized enzyme/activity of total amount of enzyme $\times 100$), you can make such statement with real scientific sound.

Without supporting data, it is simply not a reasonable statement that covalent immobilization modifies the enzyme while the chelation interaction to a metal ion not. It can occur that covalent bond formation happens far from the active site entrance and does not alter the activity (several times multipoint attachments can even enhance the stability of the enzyme). In such case, covalent immobilization can show high Y_i and Y_a as well. On the other hand, it can happen that the position of the His-tag interferes with the accessibility of the active site, therefore an IMAC immobilization can result in quite low Y_a with high Y_i .

Thus, a real comparison of immobilization methods can be based on Y_i and Y_a data but not just on speculations. If there are no comparable Y_i and Y_a data, you may state that you selected IMAC method due to its simplicity and satisfactory stability, but you cannot state that it is an optimal method for immobilization. Please clarify this issue with a short statement in the manuscript as well.

In conclusion, this work provides an alternative FMN-based flow-through cofactor regeneration system (in closed-loop mode) for FMN-dependent enzymic reductions and thus deserves publication. However, I suggest the publication of the results after minor revision addressing the residual issues indicated above.

Evaluation of the responses to Reviewer 3 by Reviewer 2:

A number of issues raised by Reviewer 3 were properly resolved. Only those responses are commented below which are not fully satisfactory.

- It is not really clear, why exactly this enzyme (TsOYE) was chosen as the obtained ees are limited to 39, 68 and 74%, which is below practical limits in the synthesis of chiral compounds. A different choice of a more selective enzyme would boost the entire manuscript significantly.

Response: The primary reason for the lower ee is the racemization of the product caused by keto-enol tautomerization in the aqueous phase during the reaction. This has been already thoroughly studied, where Fryszkowska et al (ref 57, *Advanced synthesis and catalysis*, 2009, DOI: 10.1002/adsc.200900574) have showed that the lower optical purity is due to non-enzymatic racemization. Additionally, we selected TsOYE because its thermostability makes it easy to handle and purify, allowing heat purification. High thermostability minimizes the risk of activity loss during handling. The novelty also lies in the enhanced stability achieved through coordination bond immobilization of TsOYE, a new progress towards thermophilic enzyme applications. We have

integrated the reviewer's comment by adding the sentence at the end of the introduction (page 2, highlighted in yellow): "TsOYE was coupled with SH as an oxidoreductase for its high thermostability, which minimizes activity loss during handling and allows easy heat purification."

Reaction: Authors should clarify in a more understandable manner in the main text that the lowered ee is not due to the incomplete selectivity of the enzyme but due to parallel racemization and partial reversibility of the process. (See response of Reviewer 2 on reversibility). It would be desirable to envisage (and add to the main text) possible strategies avoiding the substantial decrease of enantiomeric purity.

- It is rather unclear to the reader if the chosen (very complex) process combination results in a significant improvement over other/older existing option. A few details are mentioned in table S2 in the SI, but for the key enzyme TsOYE hardly any improvement is seen. In the main text it is mentioned as "surpasses", but only to a minor effect.

Response: The main novelty of this system lies in its use of electricity to fuel the reactions and the potential to incorporate gases other than H₂ for flavin-dependent biocatalysis. Additionally, the system's use of coordination bonds with His-tagged TsOYE for immobilization proves highly effective, leading to higher TTN. Also, Strep-tagged SH immobilized with Strep-tactin also demonstrates good compatibility with flavin-dependent reactions, whereas adsorption methods are unsuitable due to FMN cofactor adsorption. The impact of this paper resides in the combination of these findings and the fact that it can open door to other flavin-dependent reactions.

Reaction: Please add these statements (major novelty: use of electricity to fuel the reactions and the potential to incorporate gases other than H₂ for flavin-dependent biocatalysis; not a major goal: fully optimized immobilization of TsOYE, since IMAC method is simple and sufficiently stable) to the main text at proper position.

- The low Faradaic efficiency of 0.15% is rather low and the authors seem to mention an issue with the gas addition module, which indicates the reader that the entire concept may be considered non-optimal. This itself should have a significant effect on the E-factor.

Response: Thank you for this comment. We accept that optimization is needed for H₂ production and availability. The main reason for the low faradaic efficiency is that the PEM electrolyzer was commercially bought and was only able to start electrolysis of water over 3.4 V where the production of H₂ could not be decreased. Also, the gas addition module could not be closed off in the outlet because the PEM electrolyzer is unable to withstand the back pressure. Part of our project concept is the ability to easily implement any commercial electrolyzer into the gas addition module. This allows for the easy incorporation of various gases besides H₂ without requiring complex equipment such as H-cube for gas-dependent flow reactions. To incorporate reviewer's comment, we have added the following sentences in the introduction (page 2, highlighted in yellow):

"Additionally, a commercial continuous-flow hydrogenation reactor have been applied in flow biocatalysis, designed for optimizing H₂ availability which may become a relevant efficiency factor[Ref: <https://doi.org/10.3389/fceng.2021.718257>]. In this work, for the task of supplying various gases to the flow system, simple and cost-effective method by integrating gas permeable tubing in a closed loop system was designed."

Referring to the E-factor calculations, we have applied the same "simple" E-factor calculation that was used in Ramirez et al. (Frontiers in Catalysis, 2022, <https://doi.org/10.3389/fctls.2022.906694>), where solvent and H₂ were not counted. Also Zhao et al. (Chemical Science, 2021, <https://doi.org/10.1039/D1SC00295C>) and Poznansky et al. (Frontiers in Chemical Engineering, 2021, <https://doi.org/10.3389/fceng.2021.718257>) did not incorporate the whole mass of H₂ used during the experiment.

Reaction: The E-factor calculation response is reasonable, but authors should emphasize better in

the added text formulated in their response that the commercially available PEM electrolyzer was suboptimal and a better fine-tunable electrolyzer can substantially enhance this situation.

Response to Reviewers:

Reviewer #1 (Remarks to the Author):

The authors have given careful consideration to the reviewer comments and have revised the manuscript appropriately. I strongly recommend publication at this stage.

Reviewer #2 (Remarks to the Author):

The revised paper of Lim et al entitled “H₂-driven biocatalysis for flavin-dependent ene-reduction in a continuous closed-loop flow system utilizing H₂ from water electrolysis” investigates the possibility of integrating a flavin-dependent Old Yellow Enzyme (OYE) and a soluble hydrogenase to enable H₂-driven regeneration of the OYE cofactor FMNH₂—by producing H₂ with water electrolysis using a proton exchange membrane (PEM) electrolyzer—into a recirculating flow-through system via a designed gas membrane addition module at a high diffusion rate.

This work is a nice, worth-to-publish implementation of H₂-driven regeneration of the cofactor FMNH₂ in flow-through systems. Most issues which were raised by the reviewers were properly addressed. Some minor issues, however, were not fully resolved.

Notes

5) In a section (lines 213-217) of the MS it is stated: “Despite constantly attaining high conversion rate, the continuous flow reaction produced (4) with a relatively low optical purity (ee = 39% R). This decrease of optical purity was observed during the reaction time (see SI chapter 7), which is in agreement to other studies and is typically linked to racemization during sample preparations or to prolonged reaction conditions.”

Because in a non-reversible enantiotopically selective biotransformation, the degree of enantiotopic selectivity ($E = k_R/k_S$, where k_S and k_R are the pseudo first-order rate constants towards the two enantiomers) should be constant, the decreasing nature of enantiomeric excess of the product in ketoisophorone reduction (Figure S16) could also reveal reversibility of the biocatalyzed process (which could explain the decreasing equilibrium conversion with increasing substrate concentration). In this aspect it would be desirable to perform control experiments to decide the reason of decreasing ee of the product in time (e.g. put a certain amount of optically active product but without the ene-reductase into the same system and check the enantiopurity over time – this could prove chemical racemization). Please check this and discuss this issue in more detail in the main manuscript!

Response: Thank you for your detailed comment. The primary reason for the lower ee is the racemization of the product caused by keto-enol tautomerization in the aqueous phase during the reaction. This has been already thoroughly studied, where Fryszkowska et al (ref 57, *Advanced synthesis and catalysis*, 2009, DOI: 10.1002/adsc.200900574) have showed that the lower optical purity is due to non-enzymatic racemization. We have addressed the reviewer’s concern by revising the sentence in the result (page 6, highlighted in yellow): “This decrease of optical purity was observed during the reaction time (see SI chapter 7), which is in agreement to other studies and is typically linked to non-enzymatic racemization due to keto-enol tautomerization during sample preparations or to prolonged reaction conditions.”

Extended note: In the cited work [Fryszkowska et al (ref 57, *Advanced synthesis and catalysis*, 2009, DOI: 10.1002/adsc.200900574)], the Table 5 shows the ee of the results of ketoisophorone formation under the same conditions and same reaction time as a function of enzyme concentration. In fact,

the increasing enzyme concentration resulted in decreasing ee, which cannot be rationalized by the racemization (same pH, same time) but is in line with the lowering degree of reversibility (due to lower amount of enzyme). Please mention the reversibility of the reaction as another major reason of lowering the ee of the product in the discussion.

Response: We would like to thank the reviewer for this comment. To mention the enzymatic reversibility of the reaction that can cause the lower optical purity, we have modified the main text in the result section (page 4, highlighted yellow) accordingly: "Despite constantly attaining high conversion rate, the flow reaction produced (**4**) with a relatively low optical purity, i.e. from $ee = 77\%$ *R* at 2 h after substrate addition to $ee = 39\%$ *R* at 22 h (see SI chapter 7). This decrease of optical purity has been linked to non-enzymatic racemization in water due to keto-enol tautomerization during sample preparations or to prolonged reaction conditions, as well as the non-enzymatic reduction of (**3**) to racemic product (**4**) by free FMNH₂. Additionally, it has been demonstrated that a lower concentration of the enzyme contributes to higher optical purity, which introduces the possibility of reversibility of the reaction by the enzyme."

6) A preceding study (ref 49) showed that covalently bound OYEs are more stable than their EZIG-based counterparts. Your preparative results are far away from the best outcomes of other modes of OYE usage. Why did you not select the more stable mode of immobilization (which is one obvious reason of the poorer results)?

Response: Covalently bound OYEs can be more stable as in the strength of the bond compared to coordination bonds but the residual activity after immobilization is also a factor we investigated. The preceding study (ref 49) or most of the covalent bond that they use for enzyme immobilization is done by creating imine bonds with the lysine residue of enzyme molecule and the aldehyde group. Therefore, the covalent bonding process involves modifying the enzyme molecules, which can either deactivate or significantly reduce their activity after immobilization, depending on the enzyme. EziG beads utilize coordination bonds between the nickel ions and the imidazole groups of 6xHis tags attached to the protein. This approach requires no modification of the enzyme, preserving its original residual activity and offers stronger binding compared to adsorption or other non-covalent methods. Our results on the reusability of immobilized biocatalysts (Figures S7) and the upscaled experiment (Figures S8) achieving high TTN indicate that immobilization methods employing coordination bonds are highly effective for this reaction.

Extended note: Your response is not a reasonable scientific answer. If you can report (and compare) the immobilization yields (Y_i : mass of immobilized enzyme/mass of total amount of enzyme x 100) and the activity yields (Y_a : activity of immobilized enzyme/activity of total amount of enzyme x 100), you can make such statement with real scientific sound.

Without supporting data, it is simply not a reasonable statement that covalent immobilization modifies the enzyme while the chelation interaction to a metal ion not. It can occur that covalent bond formation happens far from the active site entrance and does not alter the activity (several times multipoint attachments can even enhance the stability of the enzyme). In such case, covalent immobilization can show high Y_i and Y_a as well. On the other hand, it can happen that the position of the His-tag interferes with the accessibility of the active site, therefore an IMAC immobilization can result in quite low Y_a with high Y_i .

Thus, a real comparison of immobilization methods can be based on Y_i and Y_a data but not just on speculations. If there are no comparable Y_i and Y_a data, you may state that you selected IMAC method due to its simplicity and satisfactory stability, but you cannot state that it is an optimal method for immobilization. Please clarify this issue with a short statement in the manuscript as well.

Response: We thank the reviewer for the comment. We acknowledge that covalent bond formation can also not affect the activity. In order to address the reviewer comment, we stated in the manuscript, page 3, highlighted in yellow: "This approach shows good residual activity and high immobilization yields (see SI chapter 2.4), while also simplifying the immobilization process compared to covalent bonding methods."

In conclusion, this work provides an alternative FMN-based flow-through cofactor regeneration system (in closed-loop mode) for FMN-dependent enzymic reductions and thus deserves publication. However, I suggest the publication of the results after minor revision addressing the residual issues indicated above.

Evaluation of the responses to Reviewer 3 by Reviewer 2:

A number of issues raised by Reviewer 3 were properly resolved. Only those responses are commented below which are not fully satisfactory.

- It is not really clear, why exactly this enzyme (TsOYE) was chosen as the obtained ees are limited to 39, 68 and 74%, which is below practical limits in the synthesis of chiral compounds. A different choice of a more selective enzyme would boost the entire manuscript significantly.

Response: The primary reason for the lower ee is the racemization of the product caused by keto-enol tautomerization in the aqueous phase during the reaction. This has been already thoroughly studied, where Fryszkowska et al (ref 57, *Advanced synthesis and catalysis*, 2009, DOI: 10.1002/adsc.200900574) have showed that the lower optical purity is due to non-enzymatic racemization. Additionally, we selected TsOYE because its thermostability makes it easy to handle and purify, allowing heat purification. High thermostability minimizes the risk of activity loss during handling. The novelty also lies in the enhanced stability achieved through coordination bond immobilization of TsOYE, a new progress towards thermophilic enzyme applications. We have integrated the reviewer's comment by adding the sentence at the end of the introduction (page 2, highlighted in yellow): "TsOYE was coupled with SH as an oxidoreductase for its high thermostability, which minimizes activity loss during handling and allows easy heat purification."

Reaction: Authors should clarify in a more understandable manner in the main text that the lowered ee is not due to the incomplete selectivity of the enzyme but due to parallel racemization and partial reversibility of the process. (See response of Reviewer 2 on reversibility). It would be desirable to envisage (and add to the main text) possible strategies avoiding the substantial decrease of enantiomeric purity.

Response: We incorporated the reviewer's comment about reversibility by addressing the comment from Reviewer 2 in the main manuscript. We also modified the main manuscript, page 6, highlighted yellow: "The subsequent challenge will be to incorporate immiscible organic solvents to the flow setup to increase the solubility of substrates in water and the address the poor optical purity of chiral products. TsOYE-EziG can be further investigated regarding its stability in organic solvents within a micro-aqueous environment, following previous studies on TsOYE immobilized via adsorption on Celite."

- It is rather unclear to the reader if the chosen (very complex) process combination results in a significant improvement over other/older existing option. A few details are mentioned in table S2 in the SI, but for the key enzyme TsOYE hardly any improvement is seen. In the main text it is

mentioned as “surpasses”, but only to a minor effect.

Response: The main novelty of this system lies in its use of electricity to fuel the reactions and the potential to incorporate gases other than H₂ for flavin-dependent biocatalysis. Additionally, the system's use of coordination bonds with His-tagged TsOYE for immobilization proves highly effective, leading to higher TTN. Also, Strep-tagged SH immobilized with Strep-tactin also demonstrates good compatibility with flavin-dependent reactions, whereas adsorption methods are unsuitable due to FMN cofactor adsorption. The impact of this paper resides in the combination of these findings and the fact that it can open door to other flavin-dependent reactions.

Reaction: Please add these statements (major novelty: use of electricity to fuel the reactions and the potential to incorporate gases other than H₂ for flavin-dependent biocatalysis; not a major goal: fully optimized immobilization of TsOYE, since IMAC method is simple and sufficiently stable) to the main text at proper position.

Response: To incorporate the reviewer's comment, we have clearly stated the novelty of the study by adding sentences in the manuscript page 2, highlighted yellow: “In this study, the main novelty is in development of a closed-loop flow platform for electro-driven flavin-dependent biocatalysis via H₂ as a mediator produced by a commercial PEM electrolyzer in combination with a gas addition module. Model thermostable enzyme TsOYE was newly immobilized via coordination bonds as an oxidoreductase, as a simple immobilization method and was coupled with SH. Immobilized SH via affinity was also newly applied in a flow system to allow regeneration of flavin cofactor without adsorption from the immobilization carrier.”

- The low Faradaic efficiency of 0.15% is rather low and the authors seem to mention an issue with the gas addition module, which indicates the reader that the entire concept may be considered non-optimal. This itself should have a significant effect on the E-factor.

Response: Thank you for this comment. We accept that optimization is needed for H₂ production and availability. The main reason for the low faradaic efficiency is that the PEM electrolyzer was commercially bought and was only able to start electrolysis of water over 3.4 V where the production of H₂ could not be decreased. Also, the gas addition module could not be closed off in the outlet because the PEM electrolyzer is unable to withstand the back pressure. Part of our project concept is the ability to easily implement any commercial electrolyzer into the gas addition module. This allows for the easy incorporation of various gases besides H₂ without requiring complex equipment such as H-cube for gas-dependent flow reactions. To incorporate reviewer's comment, we have added the following sentences in the introduction (page 2, highlighted in yellow):

“Additionally, a commercial continuous-flow hydrogenation reactor have been applied in flow biocatalysis, designed for optimizing H₂ availability which may become a relevant efficiency factor[Ref: <https://doi.org/10.3389/fceng.2021.718257>]. In this work, for the task of supplying various gases to the flow system, simple and cost-effective method by integrating gas permeable tubing in a closed loop system was designed.”

Referring to the E-factor calculations, we have applied the same “simple” E-factor calculation that was used in Ramirez et al. (Frontiers in Catalysis, 2022, <https://doi.org/10.3389/fctls.2022.906694>), where solvent and H₂ were not counted. Also Zhao et al. (Chemical Science, 2021, <https://doi.org/10.1039/D1SC00295C>) and Poznansky et al.(Frontiers in Chemical Engineering, 2021, <https://doi.org/10.3389/fceng.2021.718257>) did not incorporate the whole mass of H₂ used during the experiment.

Reaction: The E-factor calculation response is reasonable, but authors should emphasize better in the added text formulated in their response that the commercially available PEM electrolyzer was suboptimal and a better fine-tunable electrolyzer can substantially enhance this situation.

Response: To incorporate the reviewer's comment, we have modified the main text in the result section in page 5, highlighted yellow: "The limited electron contribution to product formation compared to other work stems from the requirement of H₂ gas outflow from the gas addition module, which is a drawback for commercially available PEM electrolyzers that has difficulty in managing back pressure if the module remained closed. Additionally, the commercial PEM electrolyzer only facilitated stable electrolysis of water at high voltage (> 3 V) to overcome the internal resistance in the cell. To increase Faradaic efficiency or the energy efficiency of the system, zero gap cells can be used where lower amount of electrical energy is used for the production of exact amount of H₂ that is needed for the biocatalytic reaction."